# Genomic Variation in Korean *japonica* Rice Varieties

**DOI:** 10.3390/genes12111749

**Published:** 2021-10-30

**Authors:** Hyeonso Ji, Yunji Shin, Chaewon Lee, Hyoja Oh, In Sun Yoon, Jeongho Baek, Young-Soon Cha, Gang-Seob Lee, Song Lim Kim, Kyung-Hwan Kim

**Affiliations:** Department of Agricultural Biotechnology, National Institute of Agricultural Sciences, Rural Development Administration (RDA), Jeonju 54874, Korea; yujishin@korea.kr (Y.S.); wowlek44@korea.kr (C.L.); hja-oh@hanmail.net (H.O.); isyoon@korea.kr (I.S.Y.); firstleon@korea.kr (J.B.); yscha63@korea.kr (Y.-S.C.); kangslee@korea.kr (G.-S.L.); greenksl5405@korea.kr (S.L.K.); biopiakim@korea.kr (K.-H.K.)

**Keywords:** *japonica*, InDel, marker, resequencing, rice, SNP, variation

## Abstract

Next-generation sequencing technologies have enabled the discovery of numerous sequence variations among closely related crop varieties. We analyzed genome resequencing data from 24 Korean temperate *japonica* rice varieties and discovered 954,233 sequence variations, including 791,121 single nucleotide polymorphisms (SNPs) and 163,112 insertions/deletions (InDels). On average, there was one variant per 391 base-pairs (bp), a variant density of 2.6 per 1 kbp. Of the InDels, 10,860 were longer than 20 bp, which enabled conversion to markers resolvable on an agarose gel. The effect of each variant on gene function was predicted using the SnpEff program. The variants were categorized into four groups according to their impact: high, moderate, low, and modifier. These groups contained 3524 (0.4%), 27,656 (2.9%), 24,875 (2.6%), and 898,178 (94.1%) variants, respectively. To test the accuracy of these data, eight InDels from a pre-harvest sprouting resistance QTL (*qPHS11*) target region, four highly polymorphic InDels, and four functional sequence variations in known agronomically important genes were selected and successfully developed into markers. These results will be useful to develop markers for marker-assisted selection, to select candidate genes in map-based cloning, and to produce efficient high-throughput genome-wide genotyping systems for Korean temperate *japonica* rice varieties.

## 1. Introduction

Rice is the world’s second most important cereal crop, following only maize (*Zea mays*). Worldwide, nearly 504 million metric tons of milled rice was produced from about 162 million hectares of paddy fields in 2019 (http://www.fao.org/faostat, accessed on 10 August 2021). Rice (*Oryza sativa*) can be classified into two main subgroups: *indica* and *japonica. Indica* genotypes are grown in tropical regions, whereas *japonica* varieties are grown in tropical or temperate regions. Generally, the genetic diversity of *japonica* varieties is lower than that of *indica* varieties [1]. Korean *japonica* rice varieties belong to the temperate *japonica* group and thus have a low level of genetic diversity. They exhibit low levels of polymorphism with traditional molecular markers, including restriction fragment length polymorphisms (RFLPs) and simple sequence repeats (SSRs), and this has hindered gene mapping and marker-assisted selection. Korean *japonica* rice varieties, however, show wide phenotypic variation in many important traits, including flowering time, plant architecture, disease and pest resistance, seed size, grain quality, pre-harvest sprouting resistance, and resistance to abiotic stress. Mapping and identification of the genes responsible for this variation and the development of selective markers are therefore required to facilitate molecular breeding.

Next-generation sequencing (NGS) technologies have revealed numerous sequence variations in closely related varieties of temperate *japonica* rice, and have enabled the development of a sufficient number of polymorphic markers to allow genotyping of populations derived from crosses between these varieties. Resequencing of the *japonica* variety Koshihikari revealed 67,051 single nucleotide polymorphisms (SNPs) relative to the reference *japonica* rice sequence, Nipponbare [2]. Moreover, 25,199 new SNPs were discovered by resequencing two other Japanese *japonica* rice varieties (Rikuu132 and Eiko), and a core set of 768 SNPs were selected for diversity and genetic analyses of biparental populations of Japanese rice accessions [3]. In addition, whole-genome sequencing of Omachi, a Japanese landrace of *japonica* rice, identified 132,462 SNPs, 16,448 insertions, and 19,318 deletions that differed between the Omachi and Nipponbare genomes [4]. Whole-genome sequencing of six cultivars (five temperate *japonica* cultivars and one tropical *japonica* cultivar (Moroberekan)) revealed that the Moroberekan genome contained five times more SNPs than the temperate *japonica* cultivars when compared with Nipponbare [5]. Whole-genome sequencing also revealed an average of 99,955 putative SNPs and 14,617 putative InDels in comparison with Nipponbare in 10 closely related rice cultivars grown in Hokkaido, the northernmost region of rice paddy cultivation in Japan [6].

Various high-throughput SNP assays have been developed in rice using SNPs discovered through resequencing. These include a custom-designed Affymetrix array consisting of 44,100 SNPs; an Illumina GoldenGate assay consisting of 1536 SNPs; and a suite of low-resolution 384-SNP assays for the Illumina BeadXpress Reader [7,8,9]. A core set of 768 SNPs were used to develop an Illumina GoldenGate platform for diversity and genetic analysis of Japanese temperate *japonica* rice varieties [2]. Two Illumina Infinium-based 6 K arrays, RiceSNP6K [10] and C6AIR [11], have been developed and used for diversity analysis, QTL mapping, marker-assisted backcross breeding (MABB), and pedigree verification of breeding lines. The C7AIR SNP array, which contains 7,098 markers, is an improved development of the previously released C6AIR [12]. The 700 K High Density Rice Array (HDRA700K) has been used for genome-wide association studies (GWAS) [13]. The 1K-Rice Custom Amplicon, or 1k-RiCA, was developed using highly informative SNPs within *indica* rice breeding pools for genetic and breeding purposes [14]. A core SNP array based on 467 Kompetitive Allele-Specific PCR (KASP) markers has been used successfully for rice germplasm assessment, genetic diversity, and population evaluation [15].

We previously analyzed genome resequencing data from 13 Korean temperate *japonica* rice varieties and discovered 740,566 SNPs, from which we developed 1225 KASP markers [16,17,18]. These markers were successfully used for QTL mapping of several important traits and MABB within Korean temperate *japonica* varieties [19,20,21,22]. However, the number of varieties analyzed in these studies was too small; thus, it was necessary to identify more of the sequence variation present in Korean temperate *japonica* rice varieties. We therefore analyzed genome resequencing data from 24 Korean temperate *japonica* rice varieties. This revealed 954,233 sequence variations consisting of 791,121 SNPs and 163,112 insertions/deletions (InDels). These results will be useful for the production of markers for marker-assisted selection, and for the development of more comprehensive and efficient high-throughput genome-wide genotyping systems for Korean temperate *japonica* rice varieties. In addition, these data provide valuable information for the development of DNA markers and the selection of candidate genes during map-based gene cloning with populations derived from crosses between Korean temperate *japonica* rice varieties.

## 2. Materials and Methods

A total of 24 Korean temperate *japonica* rice varieties (Cheongho, Dami, Dongan, Dongjin, Giho, Haechanmulgyeol, Hiami, Hwacheong, Hwayeong, Ilpum, Jinbu43, Jopyeong, Joun, Junam, Nampyeong, Odae, Saeilmi, Saenuri, Samgwang, Seogan, Seomyeong, Sindongjin, Sobi, and Unbong40) were grown in a greenhouse of the National Institute of Agricultural Sciences (NIAS) of the Rural Development Administration (RDA, Jeonju, Korea). Genomic DNA was extracted from the leaves using the DNeasy Plant Mini Kit (QIAGEN, Hilden, Germany). 

Resequencing of the entire genome of 11 varieties (Cheongho, Dami, Dongan, Haechanmulgyeol, Jinbu43, Jopyeong, Saeilmi, Seogan, Seomyeong, Sindongjin, and Sobi) was performed using an Illumina HiSeq2000 instrument (Illumina, San Diego, USA) with a paired-end library. Raw sequencing data from the remaining 13 *japonica* varieties were reported previously [16].

Analysis of the resequencing data was performed according to the methods reported by Kumagai et al. [23] using the Trimmomatic 0.36 [24], BWA-mem (v0.7.12) (https://sourceforge.net/projects/bio-bwa/, 25 August 2021), Picard 2.9.0 (http://broadinstitute.github.io/picard/, 25 August 2021), GATK (v4.1.3.0) (https://github.com/broadinstitute/gatk/, 25 August 2021), and SnpEff (v4.3t) [25] programs. Briefly, the low-quality bases and adapter sequences in each read were removed using Trimmomatic. The reads were mapped to the IRGSP-1.0 Nipponbare reference genome [26] using BWA-mem with the default setting. After removing PCR duplicates with Picard 2.9.0, the variants were called for each sample using the GATK HaplotypeCaller. The variants of each variety were combined using GATK CombineGVCFs, and varieties were genotyped using GATK GenotypeGVCFs. Hard filtering of variants was done using GATK VariantFiltration and GATK SelectVariants with the filter “QD < 5.0, FS > 50.0, SOR > 3.0, MQ < 50.0, MQRankSum < −2.5, ReadPosRankSum < −1.0, ReadPosRankSum > 3.5”. The effect of each variant site was annotated using SnpEff; rice genome annotation information from the RAP database (RAP-DB, https://rapdb.dna.affrc.go.jp/, 25 August 2021) [27] was used in the SnpEff analysis. The position, genotypes of varieties, and annotation of variants were extracted using SnpSift (v4.3t) [28]; in addition, SnpSift was used to extract variants with high and moderate impact effects. InDels longer than 20 bp were extracted using GATK SelectVariants. The nucleotide diversity (π), allele number, and frequency of alleles at each variant were calculated using vcftools (v0.1.13) [29]. The polymorphism information content (PIC) value was calculated based on the frequency of alleles.

To develop InDel markers in the *qPHS11* region (22.0–25.0 Mbp on chromosome 11), nine InDels in this region longer than 20 bp were selected, and primers were designed based on their flanking sequences using the CLC Genomics Workbench (v6.0.1) program (http://www.qiagen.com, 25 August 2021). To develop highly polymorphic InDel markers, four InDels with PIC values greater than 0.4 and without missing data were selected, and primers were designed based on their flanking sequences. In order to find sequence variations in the well-known agronomically important genes, the list of “Agronomically important genes” in RAP-DB (https://rapdb.dna.affrc.go.jp, 25 August 2021) was used. Among the found genes, four genes including *Hd1*, *Hd6*, *GS3*, and *SD1* were selected, and the primers were designed based on the flanking sequences of functional sequence variations in those genes. The primer sequences of the markers are shown in Appendix A.

Phylogenetic analysis of the 24 varieties was conducted using the SNPhylo [30] and MEGA X programs [31]. Population structure for varieties was determined using the STRUCTURE (version 2.3.4) [32,33] program, varying the number of clusters (K) from one to fifteen, with five replications. The models, following admixture and correlated allele frequency with a 5000 burnin length and a run length of 50,000, were used for conducting model-based structure analysis. Output of STRUCTURE analysis was collected using the STRUCTURE harvester [34], and the most probable K value was determined based on the LnP(D) and Evanno’s ΔK [35].

## 3. Results

### 3.1. Detection of Variations in Genome Sequences

We analyzed the genome resequencing data of 24 Korean temperate *japonica* rice varieties (Cheongho, Dami, Dongan, Dongjin, Giho, Haechanmulgyeol, Hiami, Hwacheong, Hwayeong, Ilpum, Jinbu43, Jopyeong, Joun, Junam, Nampyeong, Odae, Saeilmi, Saenuri, Samgwang, Seogan, Seomyeong, Sindongjin, Sobi, and Unbong40). The quantity of raw genome sequence data from the different varieties ranged from 14.55 Gbp (Odae) to 55.99 Gbp (Junam) with a mean of 27.50 Gbp (Appendix A). After read mapping of the Nipponbare reference genome, the mapped nucleotides ranged from 13.28 Gbp (Odae) to 52.01 Gbp (Junam) with a mean of 25.61 Gbp. The mapping depth ranged from 35.58× to 139.35× with a mean of 68.35×. We identified 954,233 sequence variants, including 791,121 SNPs and 163,112 InDels, between the 24 Korean *japonica* rice varieties. Overall, chromosome 5 contained the lowest number of variants (20,602), and chromosome 11 the highest (202,097). On average, there was one variant per 391 bp, a variant density of 2.6 per 1 kbp (Table 1). 

The distributions of sequence variations per 100 kbp interval and nucleotide diversity within a 100 kbp window over the 12 rice chromosomes are shown in Figure 1. Most intervals contained SNPs, although their density was uneven across each chromosome. Chromosomes 6, 8, 10, 11, and 12 had the widest ranges with variation density as high as 100–1000 per 100 kbp; the nucleotide diversity within a 100 kbp window was especially high over large regions of chromosomes 6, 8, and 11. By contrast, variation density and nucleotide diversity were mostly low on chromosome 5.

The distribution of InDel sizes is shown in Figure 2. InDel size ranged from 1 to 234 bp, although 1 bp InDels occurred most frequently (75,490 InDels). Further information about each InDel, including genotypes of varieties and annotation, is provided in Appendix A. InDels longer than 20 bp can be converted to markers resolvable on agarose gels, which enables their practical use in ordinary laboratories. We identified 10,860 InDels longer than 20 bp; their full details are provided in Appendix A.

To test the usefulness of the InDel data, we designed nine InDel markers in the region of *qPHS11*, a major QTL for pre-harvest sprouting resistance found in the recombinant inbred line (RIL) population derived from a cross between Odae and Unbong40 [19]. An analysis showed that eight markers revealed polymorphisms between the parental varieties, Odae and Unbong40, and one marker failed to amplify by PCR (Figure 3a). We also designed four highly polymorphic InDel markers with PIC values greater than 0.4. All of these revealed polymorphisms between the 24 varieties, as expected (Figure 3b). These results confirmed that the InDels identified in this study enabled the development of accurate and useful markers.

### 3.2. Prediction of the Effects of Variation on Gene Function

The effects of the sequence variations on gene function were predicted using the SnpEff program [25]. The impacts of the effects were categorized into four groups: high, moderate, low, and modifier. These groups contained 3524 (0.4%), 27,656 (2.9%), 24,875 (2.6%), and 898,178 (94.1%) variants, respectively (Table 2). Frameshift mutations were the most common variants in the high-impact group (2518), whereas missense mutations were the most common variants in the moderate-impact group (25,436) (Table 3). Synonymous mutations were the most common variants (19,629) in the low-impact group, whereas variations in upstream gene sequences were the most common variants (361,453) in the modifier group (Table 3). Additional information about variants with high and moderate impact is provided in Appendix A. The variation identified in this study will be very useful for selecting candidate genes in specific target regions during map-based gene cloning with populations derived from crosses between Korean temperate *japonica* varieties. 

In order to identify sequence variations in the well-known agronomically important genes, we referred to the genes in the list of “Agronomically important genes” in RAP-DB (https://rapdb.dna.affrc.go.jp, 25 August 2021), which included 73 genes. We found sequence variations in 18 genes among them (Table 4, Appendix A). Especially, the sequence variations in *SD1*, *GS3*, *HD6*, *HD3B*, *HD1*, *Hd18*, *Pia*, *Pb1*, and *Ptr* genes were identical with those that have been reported to be functional variations. Based on these results, we designed four functional markers for *Hd1*, *Hd6*, *GS3*, and *SD1* genes. A 43 bp deletion causing a frameshift in the first exon was found in *Hd1* [36,37], and was used to design a marker for *Hd1*. The genome resequencing data analysis showed that Joun and Unbong40, which are early-maturing varieties, contained the deletion genotype, while other varieties contained the reference genotype; this finding was confirmed by an analysis using the marker for *Hd1* (Figure 3c). An SNP causing the loss of the stop codon was found in *Hd6* and used to design a marker. *HD6* encodes the α-subunit of a protein kinase, CASEIN KINASE II (CK2); the Nipponbare allele contains a premature stop codon, whereas the allele found in Kasalath, an *indica* variety, does not [38]. The genome resequencing data analysis showed that only Odae contained the Kasalath allele, while all the other tested varieties had the Nipponbare allele. An analysis with the *Hd6* marker confirmed this result (Figure 3c). *GS3* regulates grain length [39]. An SNP causing a premature stop codon was found in *GS3* and was used to design a marker. The genome resequencing data analysis showed that Dami, Sindongjin, and Sobi, which are large-grained varieties, contained the premature stop codon, while the other varieties possessed the reference genotype; this finding was confirmed by an analysis using the *GS3* marker (Figure 3c). Mutations in *SD1* reduce culm length [40]. An SNP causing an amino acid change was found in this gene and used to design a marker. The genome resequencing data analysis showed that Ilpum, Jopyeong, Odae, Junam, and Seogan contained the variant *sd1* genotype, but the other varieties contained the reference genotype. An analysis with the marker for the *sd1* variant confirmed this finding (Figure 3c).

### 3.3. Structure and Phylogenetic Analysis

The SNPhylo program extracts SNP data which meet the criteria of ≥ MAF (Minor Allele Frequency) and ≤ missing rate threshold and are in approximate linkage equilibrium with each other from large SNP datasets produced by resequencing [30]. By using this program, 1758 representative SNPs were extracted with criteria of MAF higher than 0.1, missing rate lower than 0.1, and LD (Linkage Disequilibrium) threshold of 0.5 from all SNPs detected in 24 Korean *japonica* rice varieties. These SNPs were used in the following population structure and phylogeny analysis.

The population structure of 24 Korean *japonica* rice varieties was analyzed by using STRUCTURE 2.3.4 software. The Evanno’s ΔK values identified three genetically distinct populations (i.e., K = 3; Appendix A), A, B, and C (Figure 4a). Then, MEGA X programs were used to construct a phylogenetic tree of the 24 Korean *japonica* rice varieties. This analysis also divided the varieties into three groups (Figure 4b): Group A (Samgwang, Haechanmulgyeol, Junam, Seomyeong, Seogan, Hwayeong, Sobi, Dami, and Sindongjin); Group B (Dongan, Nampyeong, Dongjin, Saeilmi, Cheongho, Saenuri, Hiami, Giho, Ilpum, Hwacheong, and Joun); and Group C (Unbong40, Jopyeong, Jinbu43, and Odae).

## 4. Discussion

In this study, we identified 954,233 sequence variants, 791,121 SNPs, and 163,112 InDels. We found 50,555 new SNPs, in addition to the 740,566 SNPs reported in our previous study [16], and performed a novel analysis of InDels. This result reveals ample sequence variation in Korean *japonica* rice varieties and may explain the wide phenotypic variation observed in many important traits, including flowering time, plant architecture, disease and pest resistance, seed size, grain quality, pre-harvest sprouting resistance, and resistance to abiotic stress. Using the SnpEff program, we predicted the effect of each variant on gene function: 3524 variants were predicted to have high-impact effects as they involved frameshift mutations, the gain or loss of a stop codon, or changes at splice donor or acceptor sites. These types of variants are extremely likely to be associated with variation in phenotypic traits. Moreover, 27,656 variants were missense mutations, in-frame insertion/deletions, or 5_prime_UTR_truncation and exon_loss_variants, which are predicted to have moderate effects on function and are thus also likely to be related to phenotypic variation. In addition, we cannot exclude the possibility that the remaining variants, predicted to have a low or modifier impact on function, are related to variation in phenotypic traits. Our analysis provided the position, genotypes of tested varieties, and full annotation of each variant, including its predicted effect on function. These data will be very helpful for future map-based cloning of genes underlying important traits in populations derived from crosses between Korean *japonica* varieties. In particular, within a target region associated with a trait, candidate genes can be identified based on the presence of variants that have a high or moderate impact on gene function.

The need for high-throughput genome-wide genotyping systems using arrays is increasing rapidly as marker-assisted selection and genomic selection become more popular techniques for plant breeding [54,55]. Highly polymorphic SNPs, SNPs affecting the function of known genes controlling agronomical traits, and SNPs located within a target gene interval are commonly used to develop high-throughput genome-wide genotyping systems [10,11,14,15]. Using the information produced by this study, suitable SNPs can be easily selected to develop genotyping systems for Korean *japonica* rice varieties. This is the goal of our future research.

The development of markers resolvable on agarose gels is important for genetic research and breeding. InDels longer than 20 bp are easily visualized on agarose gels and can thus be used in ordinary laboratories without the need for expensive equipment. Shen et al. [56] identified InDels between the rice varieties Nipponbare and 9311, and used InDels of 25–50 bp to construct 108 InDel markers. A further 346 markers based on InDels of 30–55 bp were developed following a comparison of the sequences of two *indica* and one *japonica* rice reference genomes [57]. InDels longer than 20 bp were detected by positional multiple sequence alignments between wild rice species and four cultivated rice varieties, and enabled the development of 541 genome-wide markers that discriminated between alleles from cultivated rice and seven AA-genome wild rice species [58]. We identified 10,860 InDels longer than 20 bp (Appendix A), and used this information to develop and successfully use eight InDel markers in the *qPHS11* target region, as well as four other highly polymorphic InDel markers (Figure 3a,b). These results show that the InDel data generated by this study will be very useful for developing markers for fine-mapping and marker-assisted selection, as well as for the construction of a genome-wide InDel marker set for Korean *japonica* rice varieties.

Interestingly, a large difference in the numbers of sequence variations among chromosomes was observed. Chromosome 11 contained the highest number of variants (202,697), and chromosome 5 the lowest number of variants (20,602). A similar result has been reported by resequencing a Japanese temperate *japonica* rice variety, Koshihikari [2]. In comparison between Koshihikari and Nipponbare, which is the reference genome, chromosome 11 showed the highest number of SNPs (12,216), and chromosome 5 the second lowest number of SNPs (1032). Moreover, in resequencing 10 Japanese temperate *japonica* rice varieties released in Hokkaido, chromosome 11 contained the highest number of SNPs (10,870–18,779), and chromosome 5 the lowest number of SNPs (1563–2834) in comparison with Nipponbare [6]. In the study by Arai-Kichise et al. [5], six Japanese temperate *japonica* rice varieties were resequenced, and chromosome 11 contained the second highest number of SNPs (12,215–27,182), while chromosome 5 had the lowest number of SNPs (1184–6119) in comparison with Nipponbare [5]. Such a large difference seems to be seen only in temperate *japonica* rice varieties. The Moroberekan, a tropical *japonica* rice variety, did not show a large difference in SNP numbers: 61,350 on chromosome 5 and 61,169 on chromosome 11 [5]. The differences in the numbers of SNPs between chromosome 5 and chromosome 11 were much smaller in two Korean Tongil-type *indica* varieties: 72,242 and 87,759 on chromosome 5 vs. 121,783 and 126,752 on chromosome 11 [59]. The cause of the large differences in the numbers of sequence variations among chromosomes in temperate *japonica* rice varieties is unclear and needs further research. 

Overall, the genomic variation found in this study will facilitate the development of markers for mapping important genes and for marker-assisted selection, together with the development of a high-throughput genome-wide genotyping system for Korean *japonica* rice varieties. 

## 5. Conclusions

Through analyzing genome resequencing data from 24 Korean temperate *japonica* rice varieties, we discovered 954,233 sequence variations, including 791,121 SNPs and 163,112 InDels. The effect of each variant on gene function was predicted using the SnpEff program and was included in annotation. Eight InDels from a pre-harvest sprouting resistance QTL (*qPHS11*) target region, four highly polymorphic InDels, and four functional sequence variations in well-known agronomically important genes were selected and successfully developed into markers. These results will be useful to develop markers for marker-assisted selection, to select candidate genes in map-based cloning, and to produce efficient high-throughput genome-wide genotyping systems for Korean temperate *japonica* rice varieties.

## Figures and Tables

**Figure 1 genes-12-01749-f001:**
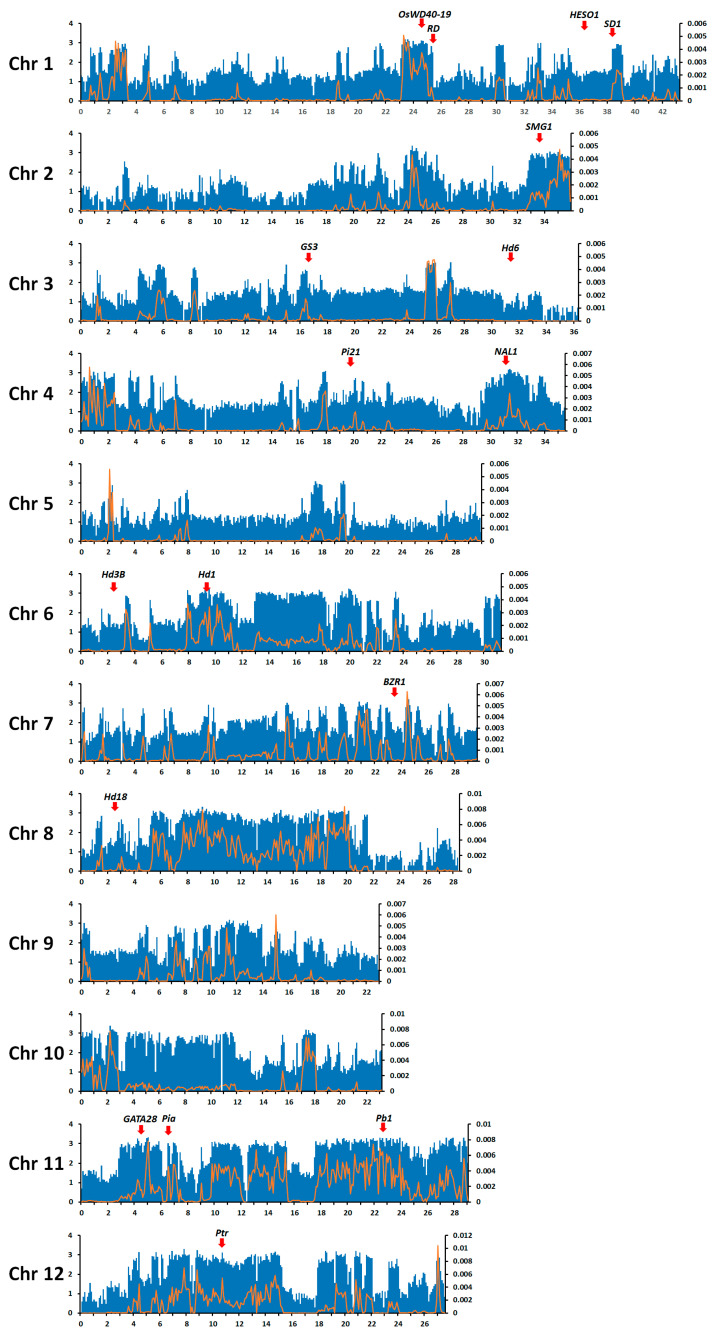
Distributions of sequence variation and nucleotide diversity per 100 kbp on each of the 12 rice chromosomes. X-axis shows the physical distance along each chromosome in mega base-pairs (Mbp). Left-hand Y-axis shows the common logarithm of the number of variations; blue bars show variation frequency. Right-hand Y-axis shows nucleotide diversity within 100 kbp windows (π), represented by the orange line. The positions of well-known agronomically important genes harboring sequence variations in 24 Korean temperate *japonica* rice varieties were indicated by red arrows.

**Figure 2 genes-12-01749-f002:**
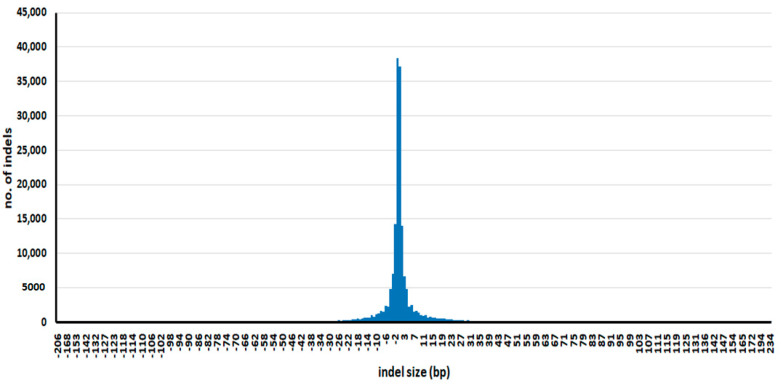
Distribution of InDel sizes. Minus values are deletions, and positive values are insertions.

**Figure 3 genes-12-01749-f003:**
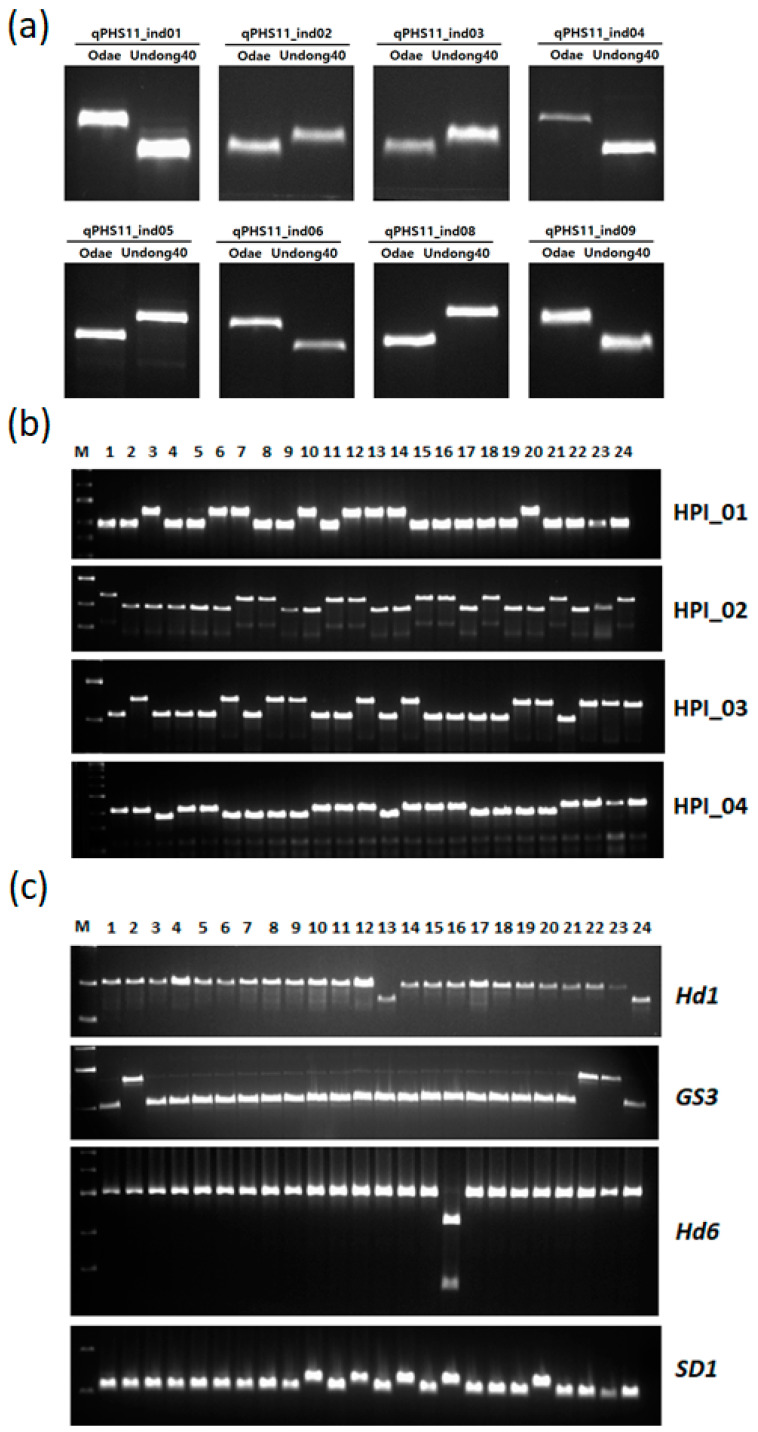
Development of markers based on sequence variation between 24 Korean *japonica* rice varieties. (**a**) Development of InDel markers in the *qPHS11* region. (**b**) Development of markers based on highly polymorphic InDels. (**c**) Development of gene-based markers; gene names are given on the right-hand side of the photograph. M: standard size markers; 1–24 represent the varieties Cheongho, Dami, Dongan, Dongjin, Giho, Haechanmulgyeol, Hiami, Hwacheong, Hwayeong, Ilpum, Jinbu43, Jopyeong, Joun, Junam, Nampyeong, Odae, Saeilmi, Saenuri, Samgwang, Seogan, Seomyeong, Sindongjin, Sobi, and Unbong40, respectively.

**Figure 4 genes-12-01749-f004:**
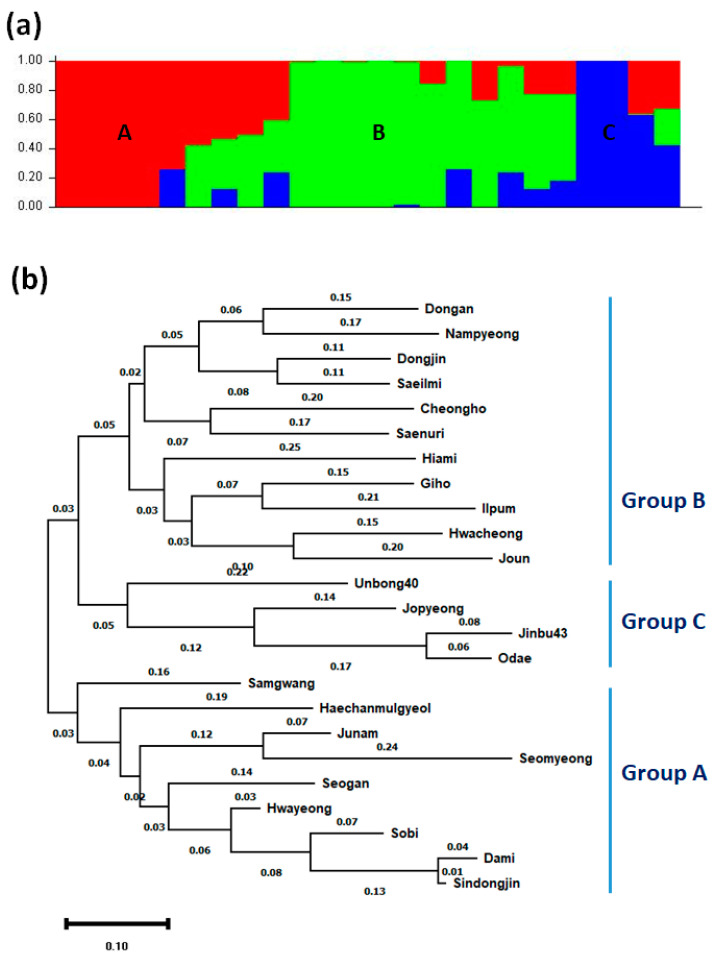
Structure and phylogeny analysis of 24 Korean temperate *japonica* rice varieties. (**a**) Assignment of 24 Korean temperate *japonica* rice varieties into three populations (A, B, and C) using STRUCTURE 2.3.4 software. The different colors represent different populations. (**b**) Phylogenetic tree of 24 Korean temperate *japonica* rice varieties. The phylogenetic tree was inferred using the maximum likelihood method and Tamura–Nei model. The tree with the highest log likelihood is shown.

**Table 1 genes-12-01749-t001:** Number of variants per chromosome.

Chromosome	No. of SNP	No. of InDels	No. of Variants ^1^	Variant Rate ^2^	Variant Density ^3^
1	51,577	12,651	64,228	673.7	1.5
2	40,194	9828	50,022	718.4	1.4
3	24,705	6669	31,374	1160.6	0.9
4	51,392	12,223	63,615	558.1	1.8
5	16,300	4302	20,602	1454.2	0.7
6	89,193	15,998	105,191	297.1	3.4
7	41,475	10,382	51,857	572.7	1.7
8	103,970	18,501	122,471	232.2	4.3
9	40,190	8938	49,128	468.4	2.1
10	74,963	13,057	88,020	263.7	3.8
11	168,447	33,650	202,097	143.6	7.0
12	88,715	16,913	105,628	260.6	3.8
All	791,121	163,112	954,233	391.1	2.6

^1^ Sum of SNPs and InDels; ^2^ mean base pair length within which a variant occurs; ^3^ mean number of variants per 1 kbp.

**Table 2 genes-12-01749-t002:** Classification of variants by predicted effects on gene function.

Chromosome	Impact of Variant Effects				Sum ^1^
	High	Moderate	Low	Modifier	
1	350	2747	2360	58,771	64,228 (6.7%)
2	254	2088	1779	45,901	50,022 (5.2%)
3	76	632	591	30,075	31,374 (3.3%)
4	310	2405	2119	58,781	63,615 (6.7%)
5	61	557	581	19,403	20,602 (2.2%)
6	238	2057	1792	101,104	105,191 (11.0%)
7	187	1618	1489	48,563	51,857 (5.4%)
8	393	2698	2530	116,850	122,471 (12.8%)
9	198	1429	1322	46,179	49,128 (5.1%)
10	266	2250	2009	83,495	88,020 (9.2%)
11	903	6906	6074	188,214	202,097 (21.2%)
12	288	2269	2229	100,842	105,628 (11.1%)
Total ^2^	3524 (0.4%)	27,656 (2.9%)	24,875 (2.6%)	898,178 (94.1%)	954,233

^1^ Number (percentage of chromosome); ^2^ number (percentage of impact).

**Table 3 genes-12-01749-t003:** Classification of variants by their effects.

Impact of SNP Effect	SNP Effect	No.
HIGH	Frameshift	2518
	Stop_gained	478
	Stop_lost	147
	Splice_acceptor_variant	143
	Splice_donor variant	127
	Start_lost	74
	Gene_fusion	34
	Feature_ablation	3
	Sum	3524
MODERATE	Missense_variant	25,436
	Inframe_insertion/deletion	2219
	5_prime_UTR_truncation&exon_loss_variant	1
	Sum	27,656
LOW	Synonymous_variant	19,629
	Splice_region_variant	3481
	5_prime_UTR_premature_start_codon_gain_variant	1736
	Stop_retained_variant	25
	initiator_codon_variant	4
	Sum	24,875
MODIFIER	upstream_gene_variant	361,453
	intergenic_region	301,015
	downstream_gene_variant	144,322
	intron_variant	48,461
	3_prime_UTR_variant	24,980
	5_prime_UTR_variant	13,281
	non_coding_transcript_exon_variant	4663
	intragenic_variant	3
	Sum	898,178

**Table 4 genes-12-01749-t004:** Summary of sequence variations in well-known agronomically important genes.

Gene Name	Gene ID	Trait	No. of Variation Sites ^1^	Reference
*OsWD40-19*	Os01g0620100	Cold tolerance	2	[41]
*RD*	Os01g0633500	Grain color	3	[42]
*HESO1*	Os01g0846450	Days to heading	1	[43]
*SD1*	Os01g0883800	Culm length	2	[40]
*SMG1*	Os02g0787300	Grain size	1	[44]
*GS3*	Os03g0407400	Grain size	1	[45]
*Hd6*	Os03g0762000	Days to heading	1	[38]
*Pi21*	Os04g0401000	Blast disease resistance	1	[46]
*NAL1*	Os04g0615000	Leaf width	1	[47]
*HD3B*	Os06g0142600	Days to heading	1	[48]
*Hd1*	Os06g0275000	Days to heading	5	[37]
*BZR1*	Os07g0580500	Plant architecture	1	[49]
*Hd18*	Os08g0143400	Days to heading	1	[50]
*GATA28*	Os11g0187200	Days to heading	4	[43]
*Pia (RGA4)*	Os11g0225100	Blast disease resistance	20	[51]
*Pia (RGA5)*	Os11g022530	Blast disease resistance	21	[51]
*Pb1*	Os11g0598500	Blast disease resistance	29	[52]
*Ptr*	Os12g0285100	Blast disease resistance	16	[53]

^1^ Number of high- or moderate-impact effect variation sites.

## Data Availability

Data is contained within the article or Appendix A.

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
