# Peer review of "Genomic Variation in Korean japonica Rice Varieties"

_genes, 2021, doi:10.3390/genes12111749_

Round 1
Reviewer 1 Report
The manuscript by Ji et al is an important, though turnkey report, which will enable marker-assisted selection and better high-throughput genome-wide genotyping systems for Korean temperate japonica rice varieties. This work builds on numerous previous studies on both japonica and indica rice varieties, following a now-standard brute-force sequencing and analysis pipeline. The methods and manuscript in general are sound, very clearly written and easy to follow. But throughout most of it, there is very little context added to the observations. The one exception to this is a very nice description on page 8 of the phenotypic effects for consequential indels in 2 known genes with effects (heading date, grain length) that segregate exactly as expected across the 24 Korean rice varieties.
I would like to see two items in the revised manuscript:
- Table 1 has the interesting observation that the variant density ranges by an order of magnitude – 0.7 / kb for Chr 5 as compared to 7.0 / kb for Chr 11. Please add a few lines in the text to speculate on why there is such a large difference. Is this fully explained by transposable elements? How well has sequence mapping been in regions of transposable elements (affected by your paired-end read length)? Is this consistent with Chromosomes 5 & 11 in other studies, e.g., of Nipponbare?
- I find it disappointing that these authors make no attempt to cross-reference their consequential and highly-variable indels to all of the proven variants with a function in the Nipponbare genotype, or to all the japonica and indica variant studies referenced in the Introduction and Discussion. This could have helped to confirm the validity of many of the variants discovered here, and also to tie many more of these discovered variants to a bona fide specific functional effect.
Author Response
I would like to express my sincere gratitude for your valuable comments.
We tried our best to incorporate your comments into our manuscript.
Reply to your comments are as follows;
1. Table 1 has the interesting observation that the variant density ranges by an order of magnitude – 0.7 / kb for Chr 5 as compared to 7.0 / kb for Chr 11. Please add a few lines in the text to speculate on why there is such a large difference. Is this fully explained by transposable elements? How well has sequence mapping been in regions of transposable elements (affected by your paired-end read length)? Is this consistent with Chromosomes 5 & 11 in other studies, e.g., of Nipponbare?
Regarding this point, I added a papagraph including description of results from other studies in lines 338-358.
2. I find it disappointing that these authors make no attempt to cross-reference their consequential and highly-variable indels to all of the proven variants with a function in the Nipponbare genotype, or to all the japonica and indica variant studies referenced in the Introduction and Discussion. This could have helped to confirm the validity of many of the variants discovered here, and also to tie many more of these discovered variants to a bona fide specific functional effect.
We extracted variants in 73 well-known agronomically important genes which were in the list of "Agronomically important genes" in RAP-DB (https://rapdb.dna.affrc.go.jp), and found sequence variations in 18 genes among them. The sequence variations in eight genes were identical with those that have been reported to be functional variations. These results were described in lines 236-241, Table 4, and Table S6.
Reviewer 2 Report
In the manuscript “Genomic Variation in Korean japonica Rice Varieties”, the authors analyzed genome resequencing data from 24 Korean temperate japonica rice varieties and discovered 954,233 sequence variations.
Although this result will facilitate the development of markers for mapping genes and for marker-assisted selection, I think that there are some important issues that must be solved by the authors first. In its current state, this paper is quite descriptive.
3.3. Phylogenetic Analysis
The authors should provide a more interpretation to explain why the varieties were divided into four groups. Structure analysis or PCA should be presented. K values should be defined to estimate the individual ancestry within the varieties.
3.2. Prediction of the Effects of Variation on Gene Function
Sequence variations in the well-known, agronomically important genes should be summarized in a figure and table instead of being described in the text.
Figure 1.
It would be informative to see if the position of agronomically important genes was plotted in Figure 1.
Table S3. Information of InDels detected in this study
What meaning did “0/0”, “1|1”, and “./.” refer to?
TableS4_Indel_S20_info.xlsx file
Tabls S3. Information about InDels longer than 20 bp detected in this study
-> Table S4. Information about InDels longer than 20 bp detected in this study
What meaning did “0/0”, “1|1”, and “./.” refer to?
I strongly encourage the deposition of the whole genome sequencing data in the International Nucleotide Sequence Database Collaboration (the DNA DataBank of Japan (DDBJ), the European Nucleotide Archive (ENA), and GenBank at NCBI).
Author Response
I would like to express my sincere gratitude for your valuable comments.
We tried to incorporate your comments into our manuscript as much as possible with all our heart.
Reply to your comments are as follows;
3.3. Phylogenetic Analysis
The authors should provide a more interpretation to explain why the varieties were divided into four groups. Structure analysis or PCA should be presented. K values should be defined to estimate the individual ancestry within the varieties.
We performed Structure analysis, and found that the tested varieties were divided into three groups. Phylogeny was also reanalyzed. These results were described in lines 270-284, Figure 4, and Figure S1. The methods used in structure analysis was added in lines 140-146.
3.2. Prediction of the Effects of Variation on Gene Function
Sequence variations in the well-known, agronomically important genes should be summarized in a figure and table instead of being described in the text.
We extracted variants in 73 well-known agronomically important genes which were in the list of "Agronomically important genes" in RAP-DB (https://rapdb.dna.affrc.go.jp), and found sequence variations in 18 genes among them. The sequence variations in eight genes were identical with those that have been reported to be functional variations. These results were described in lins 236-241, Table 4, and Table S6.
Figure 1.
It would be informative to see if the position of agronomically important genes was plotted in Figure 1.
Figure 1 was revised adding indications of the position of agronomically important genes harboring sequence variations in 24 Korean temperate japonica rice varieties.
Table S3-S4. Information of InDels detected in this study
What meaning did “0/0”, “1|1”, and “./.” refer to?
These indicate genotypes as follows; 0: reference allele, 1: the first alternative allele, 2: the second alternative allele, 3: the third alternative allele, etc.., /: not-phased genotype, |: phased genotype, ./.: missing genotype. This information was written in genotype column title in the Tables.
I strongly encourage the deposition of the whole genome sequencing data in the International Nucleotide Sequence Database Collaboration (the DNA DataBank of Japan (DDBJ), the European Nucleotide Archive (ENA), and GenBank at NCBI).
We will deposit the whole genome sequencing data of varieties in GenBank at NCBI in near future.
Round 2
Reviewer 1 Report
The authors have adequately addressed my concerns and strengthened the manuscript by better connecting it to previous work in this area.
Be sure to perform a spell and grammar check on the new text that was added as I observed a few typographical errors.
Reviewer 2 Report
The authors have responded appropriately to my concerns, providing additional data.